# Prolonged Exposition with Hyperthermic Intraperitoneal Chemotherapy (HIPEC) May Provide Survival Benefit after Cytoreductive Surgery (CRS) in Advanced Primary Epithelial Ovarian, Fallopian Tube, and Primary Peritoneal Cancer

**DOI:** 10.3390/cancers14143301

**Published:** 2022-07-06

**Authors:** Miklos Acs, Zoltan Herold, Attila Marcell Szasz, Max Mayr, Sebastian Häusler, Pompiliu Piso

**Affiliations:** 1Department of General and Visceral Surgery, Hospital Barmherzige Brüder, 93049 Regensburg, Germany; max.mayr@barmherzige-regensburg.de (M.M.); pompiliu.piso@barmherzige-regensburg.de (P.P.); 2Division of Oncology, Department of Internal Medicine and Oncology, Semmelweis University, 1083 Budapest, Hungary; herold.zoltan@med.semmelweis-univ.hu (Z.H.); szasz.attila_marcell@med.semmelweis-univ.hu (A.M.S.); 3Department of Gynecology and Gynecological Oncology, Hospital Barmherzige Brüder, 93049 Regensburg, Germany; sebastian.haeusler@barmehrzige-regensburg.de

**Keywords:** ovarian cancer, peritoneal carcinomatosis, cytoreductive surgery, HIPEC, HIPEC duration

## Abstract

**Simple Summary:**

The mainstay of therapy for primary advanced ovarian, fallopian tube, and primary peritoneal carcinoma is cytoreductive surgery (CRS) and platinum-containing chemotherapy. Since the primary and recurrent disease is mainly confined to the peritoneal cavity, intraperitoneal chemotherapy with or without hyperthermia has been evolved in order to improve survival. However, the studies to date are mainly retrospective, nonrandomized, and heterogeneous in terms of patient selection. There are currently only a limited number of reliable data on the role of hyperthermic intraperitoneal chemotherapy (HIPEC), its ideal timing of application, and duration. The aim of this study was to report our single-center experience with CRS + HIPEC involving patients with primary FIGO stage ≥ III B epithelial carcinoma. Hereby, multimodal therapy was feasible with acceptable morbidity and mortality, with no difference in overall survival between interval and upfront surgery. However, prolongation of HIPEC was shown to provide survival benefits regardless of previous administration of chemotherapy.

**Abstract:**

Background: The usage of cytoreductive surgery combined with hyperthermic intraperitoneal chemotherapy (HIPEC) for advanced gynecological cancers is increasing. Methods: Prospectively collected data of 85 advanced primary ovarian/fallopian tube cancer and peritoneal carcinoma patients of a single center were investigated. Results: A total 48, 37, 62, and 25 patients were enrolled into the HIPEC with/without neoadjuvant chemotherapy (upfront vs. interval) and into the 60 min and 90 min long HIPEC groups, respectively. Better overall survival (OS) was observed in the 90 min HIPEC group (*p* = 0.0330), compared to the 60 min HIPEC group. Neither OS (*p* = 0. 2410), disease-specific (*p* = 0. 3670), nor recurrence-free survival (*p* = 0.8240) differed between upfront and interval HIPEC. Higher peritoneal carcinomatosis index (PCI) values were associated with worse disease-specific survival (*p* = 0.0724). Age (*p* = 0.0416), body mass index (*p* = 0.0044), PCI (*p* < 0.0001), the type (*p* = 0.0016) and duration (*p* = 0.0012) of HIPEC, and increased perioperative morbidity (*p* < 0.0041) had the greatest impact on OS. Conclusions: Increasing data support the value of HIPEC in the treatment of advanced ovarian cancer. Ongoing prospective studies will definitively clarify the role and timing of this additional therapeutic approach.

## 1. Introduction

Ovarian cancer has the highest mortality rate of all gynecologic cancers and has an incidence rate of 295,000 and lethality rate of 185,000 annually worldwide [1]. Most patients (75%) are diagnosed at Fédération Internationale de Gynécologie et d’Obstétrique (FIGO) advanced stages III and IV, where the disease has spread throughout the peritoneal cavity with or without the involvement of lymph nodes or spread to distant sites [2]. The mainstay of the therapy for epithelial ovarian cancer (EOC) is primary cytoreductive surgery (CRS) followed by intravenous chemotherapy with carboplatin and paclitaxel [3]. For further prolongation of survival, CRS supplemented by hyperthermic intraperitoneal chemotherapy (HIPEC) has been introduced with increasing interest worldwide in ovarian cancer patients [4].

Since the peritoneum is the primary route of spread of EOC [5], HIPEC is supposed to provide an adjunctive approach after complete or nearly complete cytoreduction targeting remaining microscopic residual tumor intraperitoneally with increased concentration in the peritoneal cavity, thus contributing to subsequent recurrence and, consequently, to prolonged survival [6]. Despite the positive results of the first prospective randomized clinical trial (OVHIPEC 1) and numerous retrospective studies showing better outcomes with addition of HIPEC in the context of multimodal therapy [7,8,9], skepticism and disagreement are widespread worldwide as to whether CRS + HIPEC does provide additional benefit compared with CRS alone. The additional use of HIPEC found its way into national guidelines in a different manner. This procedure was included in the National Comprehensive Cancer Network (NCCN) 2019 guidelines as an optional treatment for interval-debulking surgery [10]. Furthermore, the French clinical practice guideline recommends the use of HIPEC in the case of initially nonresectable FIGO stage III ovarian, tubal, and primary peritoneal carcinoma after a complete or optimal interval surgery, which is performed after three cycles of intravenous chemotherapy [11]. While other national guidelines do not even consider this therapeutic option [12,13], some, such as in Germany, recommend the application of HIPEC only in the context of prospective randomized studies [14]. Similarly, the recently published recommendation of European Society for Medical Oncology (ESMO) and European Society of Gynecological Oncology (ESGO) consensus on ovarian cancer failed to recommend HIPEC as a standard of care for the first-line management of patients with advanced-stage epithelial ovarian cancer [3].

Despite the fact that many studies have shown prolonged survival with HIPEC therapy [4,15,16], there is also a negative, preliminary, smaller Korean prospective randomized study, which gave no benefit in terms of survival of this additional approach [17]. Thus, currently limited evidence is available from clinical studies to definitely predict the impact of HIPEC on survival in ovarian cancer patients. This is due to the fact that most studies are retrospective, nonrandomized, include both primary and recurrent cases, and also include a heterogeneous group of patients at FIGO stage. While numerous randomized prospective clinical studies are ongoing and eagerly awaited, our goal with this study is to contribute to the definitive answer to this set of questions by reporting on our 10 years of experience with CRS and HIPEC in FIGO ≥ IIIB primary epithelial ovarian, tubal, and peritoneal carcinoma. For this purpose, a clearly defined and described homogeneous collective of patients was evaluated.

The primary end point was to investigate whether the outcome is better with upfront or interval HIPEC. The secondary end point was the determination of the impact of HIPEC duration (60 vs. 90 min) on survival in a homogeneous cohort of patients.

## 2. Materials and Methods

### 2.1. Patients and Study Design

This study was performed according to STROBE requirements. From January 2011 to January 2021, a total of 85 consecutive treated patients with the histopathological diagnosis of primary epithelial ovarian, tubal and peritoneal carcinoma with FIGO stage ≥ IIIB (IIIB, IIIC and IV A and B) who received CRS and double-compound HIPEC with cisplatin and doxorubicin were reviewed and included. All patients who had previous surgery due to peritoneal metastases, recurrence, borderline tumor or other histological subtypes were excluded.

The clinicopathological data had been prospectively entered in the national HIPEC registry administered by the German Society for General and Visceral Surgery (DGAV) and was retrospectively analyzed for this study. All the patients had agreed to data recording in the registry and simultaneously to use their anonymized data for quality assurance and research purposes by written and verbal informed consent prior to surgery. Therefore, and due to the retrospective nature of this study, no institutional or further review board approval was necessary. All patients were treated according to multidisciplinary recommendations.

### 2.2. Details of CRS + HIPEC

A closed HIPEC with a goal temperature of 42 °C with intraperitoneal chemotherapy with cisplatin 75 mg/m^2^ of body surface area and doxorubicin 15 mg/m^2^ of body surface area was administered immediately after cytoreductive surgery for 60 or 90 min. The HIPEC treatment duration increased from 60 min to 90 min in 2018 due to an institutional protocol change in accordance with van Driel′s study [9]. Cisplatin and doxorubicin were added to a 3000 to 4000 mL isotonic saline solution in accordance with the body surface area of the patients. The mean flow rate was 1400–1800 mL/minute. The global amount of perfusion with the cytotoxic agents was 4000 mL. During HIPEC treatments, temperatures were monitored in the right subphrenic and pelvic area.

The surgeries were performed by the same surgical team extended by a gynecological oncologist (ovarian team). The same members of the ovarian team were involved in the multidisciplinary patient selection. The extent of peritoneal dissemination was assessed preoperatively using abdominal and chest CT scan. Clinicopathological data were obtained from prospectively collected database and electronic medical reports. All the patients were staged based on the 2014 International FIGO staging system [18]. Neoadjuvant chemotherapy (NACT) prior interval CRS was considered for patients who were not good candidates for primary CRS due to frailty, poor performance status, comorbidities, or who had disease unlikely to become completely cytoreduced or to become a minimally residual disease at the time of presentation, as has been suggested by the Society of Gynecologic Oncology (SGO) and American Society of Clinical Oncology (ASCO) [19]. In patients in whom the possibility of complete cytoreduction was not clear, we obtained diagnostic laparoscopy to determine resectability. The laparoscopy was repeated with still unclear findings at the re-staging after NACT. Patients who received neoadjuvant chemotherapy, 3–7 cycles of carboplatin (area under the curve of 5 mg per milliliter per minute, AUC 5) and 175 mg/m^2^ paclitaxel were administered intravenously. CRS + HIPEC was categorized as upfront or interval if performed as the first surgical treatment or after neoadjuvant chemotherapy, respectively. The completeness of cytoreduction (CC) was scored as proposed by Sugarbaker: CC-0: no residual disease; CC-1: residual nodules measuring less than 2.5 mm; CC-2: residual nodules measuring between 2.5 mm and 2.5 cm; and CC-3: residual nodules greater than 2.5 cm [20]. The extent of peritoneal disease was assessed by using the peritoneal carcinomatosis index (PCI), which ranges from 1 to 39 [20].

### 2.3. Clinical Characteristics

Several variables (pre-surgical, surgical and postoperative features) were analyzed. Postoperative adverse events were categorized according to the Clavien–Dindo Classification, and major complication was defined as Grade ≥ III [21]. Recurrence-free (RFS), disease-specific (DSS) and overall survival (OS) were calculated from the date of surgery (CRS + HIPEC) to the date of recurrence/progression, cancer-related death and death from any cause, respectively. For non-deceased, non-relapsed patients, the time interval between surgery and last follow-up date was chosen. Follow-up of patients was terminated at 31-DEC-2021.

### 2.4. Statistical Analysis

Statistical analysis was performed with R version 4.2.0 (R Core Team, 2022, Vienna, Austria). Welch’s *t*-test, Wilcoxon rank sum test and Fisher’s exact test were used for group comparisons between cohorts. Patient survival was evaluated using competing risk Cox regression models (R package “survival”, Therneau and Grambsch, version 3.3-1). The R package “car” (Fox and Weisberg, version 3.0-13) was used to calculate partial likelihood ratio or Wald tests for the Type II analysis of deviance tables. *p* < 0.05 was considered statistically significant and *p*-values were corrected with the Holm method [22] for multiple-comparisons problem. Continuous, count, and survival data were expressed as mean ± standard deviation, the number of observations (percentage), and as hazard ratio (HR) with 95% confidence interval (95% CI), respectively. Survival curves were drawn with the R-package “survminer” (Kassambara, Kosinski and Biecek, version 0.4.9).

## 3. Results

A total of 85 primary ovarian cancer patients, who underwent CRS + HIPEC, were included in the study. The study subjects were divided into the following groups based on two different factors. First, 37 and 48 patients were enrolled into the ‘patients not receiving cisplatin/paclitaxel NACT group’ (upfront HIPEC) and into the ‘patients receiving cisplatin/paclitaxel NACT group’ (interval HIPEC), respectively. Second, patients were grouped based on the duration of HIPEC: 60 patients were enrolled into the ‘60 min HIPEC’ group and 25 into the ‘90 min HIPEC’ group. Pre-, peri- and postoperative features of patients are summarized in Table 1. The comparisons between upfront and interval HIPEC and between 60 and 90 min HIPEC were carried out separately, and no further subgroup analyses were performed.

Relapse/progression occurred in 23 cases, and three separate endpoint events were defined for DSS survival analyses: death related to cancer and to postoperative complications (PC) and lost-to-follow-up (LFU). A total of 43 and 2 cancer and PC-related deaths were observed, and 3 and 37 patients were LFU and alive at the end of our observation. The two PC-related deaths occurred due to the following. One patient died on the 12th postoperative day due to renal insufficiency and bilateral pneumonia with consecutive septic–toxic multiorgan failure. The second patient died on the 10th postoperative day. She required reoperation due to duodenal ulcer perforation with 4-quadrant peritonitis on the 4th postoperative day. After the reoperation, the patient aspirated, and circulatory arrest occurred leading to hypoxic brain damage. The median follow up was 42.6 months. The Kaplan–Meier 3-year overall survival (OS) rate was 53.7% for the whole cohort, 46.7% for 60 min and 77.5% for 90 min HIPEC, respectively.

### 3.1. Interval vs. Upfront HIPEC

Forty-eight and thirty-seven patients underwent interval and upfront HIPEC, respectively. PCI was significantly lower in the interval HIPEC group (*p* < 0.0001, Table 1). FIGO stage IV (crude *p* = 0.0025) and the presence of hepatic and/or splenic parenchymal metastasis (crude *p* = 0.0053) was more common in the upfront HIPEC group. Six of nineteen (31.6%) FIGO IV patients received NACT. In those patients in whom complete cytoreduction appeared feasible during preoperative diagnostics and intraoperative assessment, NACT was omitted and primary resection was preferred. Marginally shorter operation time (excl. HIPEC) was required (*p* = 0.0892; crude *p* = 0.0015), and the following procedures were less characteristic for the interval HIPEC group: colon (*p* = 0.0055) and rectosigmoid resection (*p* = 0.0260), protective and/or long-term ileostomy (crude *p* = 0.0159) right upper-quadrant peritonectomy (crude *p* = 0.0345) and stripping of the omental bursa (crude *p* = 0.0317). No differences in the other pre-, peri- and postoperative features were found (Table 1). Neither DSS (*p* = 0.3670), OS (*p* = 0.2410), nor RFS (*p* = 0.8240) was affected by the type of HIPEC. It must be highlighted, however, that patients of the interval HIPEC groups seemed to have worse survival, instead within the middle of the survival curves (Figure 1).

### 3.2. Does the Duration of HIPEC Affect Clinical Parameters and Patient Survival?

Administration of NACT was more common prior 90min-long HIPEC treatments (48.3% vs. 76%; crude *p* = 0.0298). PCI (*p* = 0.0593, crude *p* = 0.0009) and the duration of the tumor removal surgery excl. HIPEC (crude *p* = 0.0145) was higher and longer in 60 min long HIPEC, respectively. The following peri- and postoperative differences were found between the two groups. Colon (crude *p* = 0. 0097) and rectosigmoid resection (crude *p* = 0.0091), protective and/or long-term ileostomy (crude *p* = 0.0080), and left upper-quadrant peritonectomy (crude *p* = 0.0149) was needed more often in the 60min-long HIPEC group. Average ICU stay was 1.5 day shorter in the case of patients belonging to the 90min-long HIPEC group (crude *p* = 0.0361), and they needed fewer units of erythrocyte (crude *p* = 0.0511) and/or albumin concentrates (*p* = 0.0015).

Better OS was observed for the patients belonging to the 90min-long HIPEC group (HR: 0.3225, 95% CI: 0.1140–0.9126, *p* = 0.0330). However, only marginal and non-significant differences were found in the case of DSS (HR: 0.3895, 95% CI: 0.1494–1.0160, *p* = 0.0538) and RFS (HR: 0.4640, 95% CI: 0.1408–1.5290, *p* = 0.2070), respectively (Figure 2). The effect of NACT over survival during the different HIPEC durations was also analyzed (Figure 3). Compared to the 60 min HIPEC without NACT subgroup, marginally better OS (HR: 0.5683, 95% CI: 0.3110–1.0380, *p* = 0.0661, Figure 3A) and DSS (HR: 0.6292, 95% CI: 0.3325–1.1910, *p* = 0.1550) were found for the patients receiving NACT prior to the 60 min HIPEC, while there was no difference in RFS (*p* = 0.5000). The use of NACT had no effect on the survival of patients within the 90 min HIPEC group (OS: *p* = 0.9410; DSS: *p* = 0.9360; RFS: *p* = 0.9720; Figure 3B).

### 3.3. Analysis of Other Parameters on Patient Survival Data

Another interest of our research was whether previous, incomplete tumor removal surgeries would affect the outcome of CRS + HIPEC. A total of 31 of the 85 (36.47%) patients had at least one previous, incomplete tumor removal surgery with an average of 134 ± 79 days interval between the previous and CRS + HIPEC surgery. Marginally better DSS was observed for patients with longer interval between the two operations (HR: 0.8641, 95% CI: 0.7348–1.0160, *p* = 0.0774), while no statistical difference was found in the case of OS (*p* = 0.2270) and RFS (*p* = 0.7330). Similarly, no difference was observed between the survival of patients with and without previous surgery (DSS: *p* = 0.4300; OS: *p* = 0.4290; RFS: *p* = 0.8740; Figure 4). Adjustment in the previously presented survival models for previous, incomplete tumor removal surgeries did not affect their result.

The effects of various clinical characteristics on patient survival were analyzed using uni- and multivariate survival models. Similar to previous findings [23,24,25,26], age, body mass index, FIGO stage, ASA score, CC score and Clavien–Dindo complication grade had a significant effect on DSS, OS and/or RFS (Table 2). It has to be highlighted that based on our multivariate survival results, the duration and type of HIPEC has more importance in the case of DSS and OS than CC score or FIGO stage. Neither of the two parameters had any effect on RFS (Table 2).

## 4. Discussion

In the natural history of ovarian cancer, the disease is confined to the peritoneal cavity in the vast majority of patients at initial diagnosis and in recurrence [27]. This prerequisite creates an ideal target for intraperitoneal (IP) chemotherapy, which leads to high peritoneal to plasma ratios compared with intravenous (IV) treatment [28]. In the meantime, multiple randomized trials and subsequent meta-analyses support the superiority of IV/IP over IV treatment in optimally cytoreduced patients [29,30,31]. However, secondary to toxicity and the difficulty at the administration of IP therapy, less than half of eligible patients received the treatment in the report by Wright et al. [31]. In order to avoid the difficulties caused by the intraperitoneal port catheter and to take advantage of the additional well-established effects of HIPEC with cisplatin (treating microscopical residual disease, due to the “peritoneal plasma barrier” providing dose-intensive therapy in the peritoneal cavity, increasing drug penetration into the tissues, and increasing the cytotoxicity of cisplatin through a synergic effect with hyperthermia [4,32,33]), it seems reasonable to optimize the parameters in the context of HIPEC. In the absence of a standardized protocol for delivering HIPEC in ovarian cancer due to the many possible variables (dosage, duration of perfusion, and methodologies), there is a wide range of variation in each parameter worldwide [34]. Indeed, to increase the number of intraperitoneal chemotherapy cycles with or without hyperthermia is proving difficult for technical reasons; nevertheless, to extend the treatment duration seems reasonable. In addition, to establish an effective protocol, it is necessary to test the parameters one by one in vivo, which we measured in the present study by comparing the duration of HIPEC in 60 versus 90 min. Hereby, we observed prolonged survival for the patients belonging to the 90 min HIPEC group. Likewise, in a recent meta-analysis by Wang and associates [35], no differences were found in subgroup analyses in OS and disease-free survival (DFS) in patients who received HIPEC for 30 min compared with the CRS group. Moreover, patients who received HIPEC for 60 min or 90 min exhibited significantly improved OS (HR = 0.47, 95% CI = 0.29–0.78, *p* < 0.01 and HR = 0.59, 95% CI = 0.40–0.88, *p* < 0.01, respectively); furthermore, in the case of 90 min treatment, DFS also significantly improved (HR = 0.62, 95% CI = 0.47–0.81, *p* < 0.01) [35]. Considering the safety of prolonged therapy, it has been demonstrated by our group in a comparative study that 90 min cisplatin and doxorubicin HIPEC administration did not increase perioperative morbidity and mortality compared to the 60 min administration, with simultaneous prolongation of exposure time and potential cytotoxic drug effect [36]. In the latter study, 35 of 120 (29.2%) patients had had ovarian cancer, out of which those with primary epithelial diagnosis were also included in the current study [36]. Summarizing the clinical and external results including the randomized prospective studies to date, our protocol changed in the second half of 2021 to a 90 min cisplatin monotherapy, which is the most effective HIPEC therapy currently known to treat EOC.

The optimal timing of CRS + HIPEC has been investigated also by other authors [7], but it has not been widely established yet. After the encouraging result of van Driel and colleagues [9] in the interval setting, interest has turned to this timing. We failed to present the superiority of interval HIPEC over upfront HIPEC in our patient cohort with FIGO ≥ IIIB stage. This finding is in line with Spiliotis et al. [7] who reported an OS for upfront CRS + HIPEC of 48 months, and for interval CRS + HIPEC 30 months (*p* = 0.074) in a similar cohort of patients, with the addition of FIGO stage IIIA compared to our study [7].

Even though NACT is increasingly used as the primary treatment for EOC [37], controversies persist on whether NACT might act as a driver for chemotherapy resistance [38,39]. Based on two randomized controlled trials without HIPEC, survival after upfront cytoreduction or interval cytoreduction with three cycles of carboplatin and paclitaxel was similar [40,41]. The many known advantages of NACT, such as reducing the extent of surgery and the need for multivisceral resections; increasing the rate of complete resection; and selection of patients who are responsive to chemotherapy and thus selecting the appropriate candidates for CRS, have been demonstrated several times by other authors and in the current study as well [42,43]. However, another aspect is the high incidence of occult microscopic disease and scar tissues after neoadjuvant chemotherapy, which may involve tumor tissue and which, according to the literature, may occur in half of the cases [44,45]. For this reason, some authors recommend routine total parietal peritonectomy regardless of visible disease [46]. In our opinion, and based on clinical experience, the strict three cycles of neoadjuvant treatment before cytoreductive surgery is not always feasible, since in many patients further neoadjuvant cycles only lead to the patient receiving complete macroscopic or nearly complete cytoreduction by reducing the number of cycles (assuming a response) of peritoneal disease. This opinion is also shared by others [42,46]. Conversely, others have reported poorer prognosis with prolonged neoadjuvant chemotherapy (after more than four cycles of NACT) compared to those who were operated on earlier [47]. We cannot share this opinion, as our results showed no survival difference after multiple neoadjuvant cycles.

It is still unclear to what extent the above-mentioned occult remaining tumor tissue and chemotherapy resistance play a role. Despite the fact that many authors describe that complete tumor clearance after neoadjuvant chemotherapy is higher than after upfront cytoreduction [4,43,48], as well as known residual tumor being one of the most important prognostic factors [49], it would be logical that this difference could be reflected in survival, but this failed to show survival benefit after neoadjuvant chemotherapy with or without HIPEC [7,40]. This also means that there is currently an unclear clinical difference and impact of residual tumor tissue after upfront and interval cytoreduction [43]. In the future, the recently developed novel fluorescent drug pafolacianine (Cytalux) may serve intraoperatively to bring occult lesions especially in the interval setting to visual inspection, thus facilitating macroscopic complete tumor resection [50].

The debate around neoadjuvant chemotherapy is to be decided by the ongoing prospective randomized clinical trial TRUST [51]. Nevertheless, registering that the survival of the upfront and interval groups is similar, and less aggressive surgeries were performed, with fewer ileostomies in patients treated with NACT, raises the possible need to re-evaluate the SGO and ASCO schedules in clinical practice in relation to neoadjuvant chemotherapy. Lastly, it should be noted that Liu et al. [52] reported that more than a quarter of neoadjuvantly treated advanced ovarian cancer patients will never reach the point to undergo cytoreductive surgery, and these patients have a significantly increased all-cause mortality compared to those who have undergone surgery at some point during their disease course. In the event that the role of HIPEC is supported by randomized prospective studies and widely accepted as part of the therapeutic protocol, a further question remains how this adjunctive therapy can be integrated into the multimodal treatment of ovarian cancer, in particular systemic chemotherapy, antibody and novel rapidly evolving targeted therapies and biological agents.

### Limitations of the Study

The limitations of this study should be addressed. Due to its retrospective nature, the present study may contain biases. It is worth noting that some of the patients enrolled in the study were referred to our tertiary center, often with extensive peritoneal carcinomatosis with more complex cases, and that the number of cycles of prior neoadjuvant chemotherapy was administered by external oncologists, a decision on which we had no influence. Furthermore, no BRCA (breast cancer gene) status was specified, which may even play a role in survival. The lack of a control group precludes a reliable conclusion on the benefit of HIPEC. Due to the shortness of the follow-up period in the 90 min HIPEC group, further observation is needed. Further possible inherent confounders are the selection biases associated with the retrospective nature of the data and the relatively small number of cases. Furthermore, the current study is a 10-year retrospective series where the duration of HIPEC has changed; however, all other variables including the surgical team have remained unchanged.

## 5. Conclusions

Growing clinical data suggest that the addition of HIPEC after CRS for primary ovarian cancer patients is an additive therapy well established in vitro and in vivo that can further improve overall survival. Ongoing prospective studies should determine the best timing of this treatment and predict the definitive effect on survival. In this study we report the superiority of prolonged application of HIPEC. To further evaluate this effect, longer observation time is necessary.

## Figures and Tables

**Figure 1 cancers-14-03301-f001:**
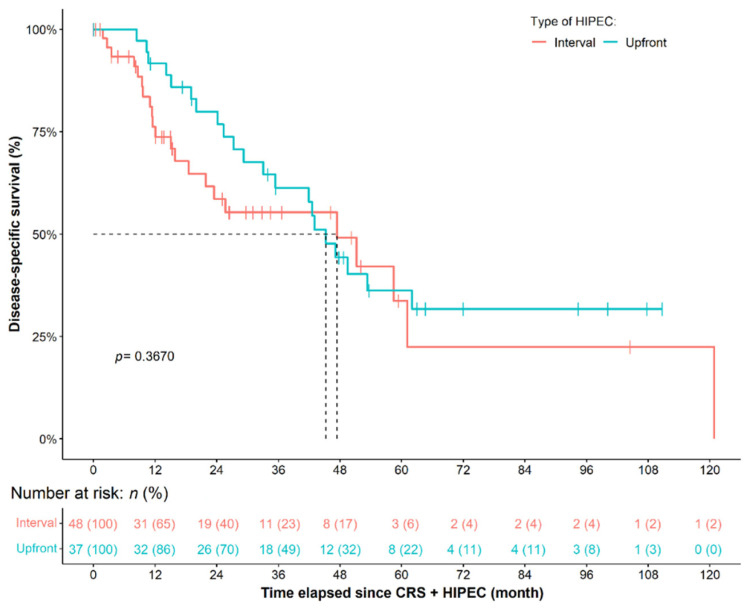
Primary ovarian cancer patients, who needed neoadjuvant chemotherapy (NACT) prior to CRS + HIPEC (interval HIPEC) had seemingly worse disease-specific survival compared to those who did not need NACT (upfront HIPEC). However, no statistical difference could be justified due to the change seen in the survival curve of interval HIPEC patients within the middle of the observation period. CRS: cytoreductive surgery; HIPEC: hyperthermic intraperitoneal chemotherapy.

**Figure 2 cancers-14-03301-f002:**
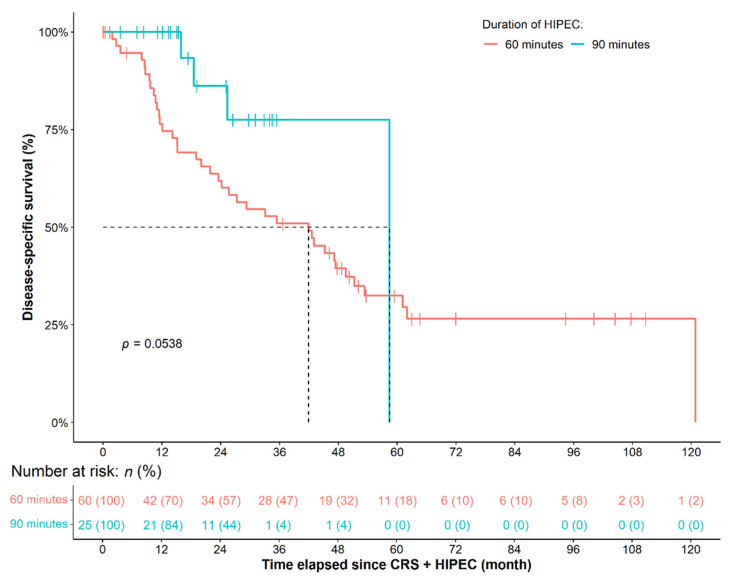
Survival of primary ovarian cancer patients, who received 60 or 90 min-long HIPEC treatment. CRS: cytoreductive surgery; HIPEC: hyperthermic intraperitoneal chemotherapy.

**Figure 3 cancers-14-03301-f003:**
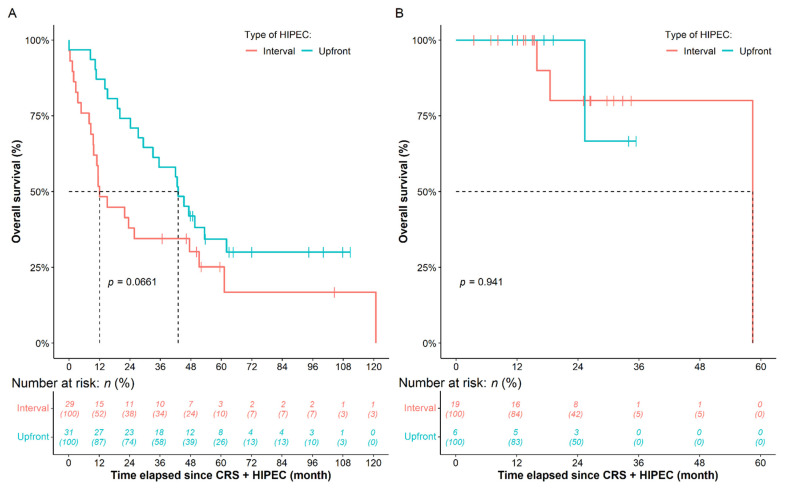
Survival difference of patients receiving neoadjuvant chemotherapy (NACT) prior to the (**A**) 60 min and (**B**) 90min-long HIPEC surgery, compared to those patients who did not receive NACT. CRS: cytoreductive surgery; HIPEC: hyperthermic intraperitoneal chemotherapy.

**Figure 4 cancers-14-03301-f004:**
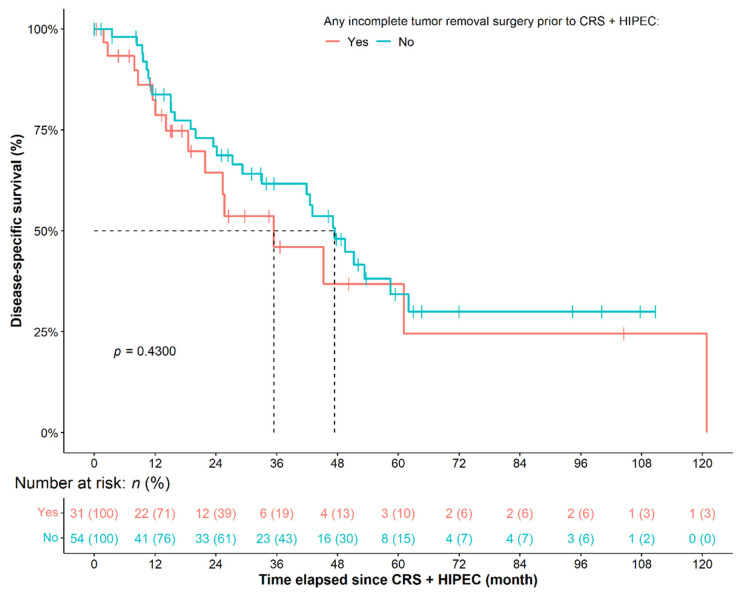
Survival of primary ovarian cancer patients, who had or did not have any previos, incomplete tumor removal surgery prior to the CRS + HIPEC procedure. CRS: cytoreductive surgery; HIPEC: hyperthermic intraperitoneal chemotherapy.

**Table 1 cancers-14-03301-t001:** Pre-, peri- and postoperative demographic and clinical characteristics of study participants. Unit of frequency data is the number of observations (percentage).

Clinical Characteristics	Total (*n* = 85)	Upfront HIPEC (*n* = 37)	Interval HIPEC (*n* = 48)	Crude *p*-Value	*p*-Value	60 min HIPEC (*n* = 60)	90 min HIPEC (*n* = 25)	Crude *p*-Value	*p*-Value
Age (year)	62.16 ± 10.29	61.98 ± 12.33	62.52 ± 8.57		1.0000	62.50 ± 10.87	61.77 ± 9.01		1.0000
Body mass index (kg/m^2^)	26.28 ± 5.39	26.74 ± 5.51	25.91 ± 5.29		1.0000	26.12 ± 5.00	26.62 ± 6.28		1.0000
ASA score					1.0000				1.0000
- I–II	20 (23.53%)	9 (24.32%)	11 (22.92%)			13 (21.67%)	7 (28%)		
- III–IV	65 (76.47%)	28 (75.68%)	37 (77.08%)			47 (78.33%)	18 (72%)		
Histology					1.0000				1.0000
- HGS ovarian, fallopian tube	61 (71.76%)	27 (72.97%)	34 (70.83%)			42 (70.00%)	19 (76%)		
- HGS primary peritoneal	21 (24.71%)	9 (24.32%)	12 (25%)			15 (25.00%)	6 (24%)		
- Endometrioid	1 (1.18%)	0 (0%)	1 (2.08%)			1 (1.67%)	0 (0%)		
- Mucinous	1 (1.18%)	1 (2.7%)	0 (0%)			1 (1.67%)	0 (0%)		
- Malignant Brenner tumor	1 (1.18%)	0 (0%)	1 (2.08%)			1 (1.67%)	0 (0%)		
FIGO Stage				0.0025	0.1496				1.0000
- IIIB	11 (12.94%)	5 (13.51%)	6 (12.5%)			9 (15.00%)	2 (8%)		
- IIIC	55 (64.71%)	19 (51.35%)	36 (75%)			36 (60.00%)	19 (76%)		
- IVA	3 (3.53%)	0 (0%)	3 (6.25%)			3 (5.00%)	0 (0%)		
- IVB	16 (18.82%)	13 (35.14%)	3 (6.25%)			12 (20.00%)	4 (16%)		
Cause of FIGO IV									
- Abdominal wall met.	7 (8.24%)	5 (13.51%)	2 (4.17%)		1.0000	4 (6.67%)	3 (12%)		1.0000
- Cardiophrenic lymph node met.	2 (2.35%)	1 (2.7%)	1 (2.08%)		1.0000	2 (3.33%)	0 (0%)		1.0000
- Hepatic and/or splenic parenchymal met.	6 (7.06%)	6 (16.22%)	0 (0%)	0.0053	0.3136	6 (10.00%)	0 (0%)		1.0000
- Inguinal lymph node met.	1 (1.18%)	1 (2.7%)	0 (0%)		1.0000	1 (1.67%)	0 (0%)		1.0000
- Malignant pleural effusion	3 (3.53%)	1 (2.7%)	2 (4.17%)		1.0000	3 (5.00%)	0 (0%)		1.0000
- Met. of the intestinal mucosa	1 (1.18%)	1 (2.7%)	0 (0%)		1.0000	0 (0%)	1 (4%)		1.0000
- Pleural carcinosis	1 (1.18%)	0 (0%)	1 (2.08%)		1.0000	1 (1.67%)	0 (0%)		1.0000
Cycles of carboplatin/TAX NACT					–			0.1273	1.0000
- 2–3	17 (20%)	0 (0%)	17 (35.42%)			9 (15.00%)	8 (32%)		
- 4–5	9 (10.59%)	0 (0%)	9 (18.75%)			7 (11.67%)	2 (8%)		
- 6–7	22 (25.88%)	0 (0%)	22 (45.83%)			13 (21.67%)	9 (36%)		
Peritoneal carcinomatosis index	11.41 ± 6.45	15.00 ± 4.96	8.60 ± 5.79		<0.0001	12.92 ± 5.65	7.72 ± 6.29	0.0009	0.0593
CC score					1.0000				1.0000
- CC-0	67 (78.82%)	30 (81.08%)	37 (77.08%)			44 (73.33%)	23 (92%)		
- CC-1	16 (18.82%)	6 (16.22%)	10 (20.83%)			14 (23.33%)	2 (8%)		
- CC-2	2 (2.35%)	1 (2.7%)	1 (2.08%)			2 (3.33%)	0 (0%)		
Cisplatin (mg)	132 ± 16.46	134.34 ± 17.37	130.20 ± 15.67		1.0000	130.69 ± 14.75	135.16 ± 19.97		1.0000
Doxorubicin (mg)	26.58 ± 3.10	27.01 ± 3.17	26.11 ± 2.91		1.0000	26.28 ± 2.54	27.04 ± 4.00		1.0000
Duration of HIPEC (min)				0.0298	1.0000				–
- 60	60 (70.59%)	31 (83.78%)	29 (60.42%)			–	–		
- 90	25 (29.41%)	6 (16.22%)	19 (39.58%)			–	–		
Type of HIPEC					–			0.0298	1.0000
- Upfront	37 (43.53%)	–	–			31 (51.67%)	6 (24%)		
- Interval	48 (56.47%)	–	–			29 (48.33%)	19 (76%)		
Length of surgery excl. HIPEC (min)	317 ± 94	353 ± 77	290.38 ± 97.59	0.0015	0.0892	336.87 ± 74.26	270.80 ± 118.43	0.0145	0.8676
Surgical procedures									
- Perit.: parietal	74 (87.06%)	34 (91.89%)	40 (83.33%)		1.0000	52 (86.67%)	22 (88%)		1.0000
- Perit.: pelvis	74 (87.06%)	31 (83.78%)	43 (89.58%)		1.0000	54 (90.00%)	20 (80%)		1.0000
- Perit.: omental bursa	24 (28.24%)	15 (40.54%)	9 (18.75%)	0.0317	1.0000	15 (25.00%)	9 (36%)		1.0000
- Perit.: right upper quadrant	58 (68.24%)	30 (81.08%)	28 (58.33%)	0.0345	1.0000	41 (68.33%)	17 (68%)		1.0000
- Perit.: left upper quadrant	35 (41.18%)	19 (51.35%)	16 (33.33%)		1.0000	30 (50%)	5 (20%)	0.0149	0.8814
- Full-thickness resection	12 (14.12%)	4 (10.81%)	8 (16.67%)		1.0000	9 (15.00%)	3 (12%)		1.0000
- Ana.: small bowel–small bowel	7 (8.24%)	3 (8.11%)	4 (8.33%)		1.0000	5 (8.33%)	2 (8%)		1.0000
- Ana.: stomach–small bowel	2 (2.35%)	0 (0%)	2 (4.17%)		1.0000	2 (3.33%)	0 (0%)		1.0000
- Ana.: small bowel–colon	25 (29.41%)	14 (37.84%)	11 (22.92%)		1.0000	19 (31.67%)	6 (24%)		1.0000
- Ana.: colon–colon	3 (3.53%)	3 (8.11%)	0 (0%)		1.0000	3 (5.00%)	0 (0%)		1.0000
- Ana.: colon–rectum	39 (45.88%)	21 (56.76%)	18 (37.50%)		1.0000	29 (48.33%)	10 (40%)		1.0000
- Ana.: small bowel–rectum	7 (8.24%)	4 (10.81%)	3 (6.25%)		1.0000	5 (8.33%)	2 (8%)		1.0000
- Colostomy	3 (3.53%)	3 (8.11%)	0 (0%)		1.0000	3 (5.00%)	0 (0%)		1.0000
- Ileostomy	25 (29.41%)	16 (43.24%)	9 (18.75%)	0.0159	0.9220	23 (38.33%)	2 (8%)	0.0080	0.5015
- Colon resection	46 (54.12%)	29 (78.38%)	17 (35.42%)		0.0055	38 (63.33%)	8 (32%)	0.0097	0.5947
- Small bowel resection	15 (17.65%)	7 (18.92%)	8 (16.67%)		1.0000	11 (18.33%)	4 (16%)		1.0000
- Rectosigmoid resection	43 (50.59%)	27 (72.97%)	16 (33.33%)		0.0260	36 (60.00%)	7 (28%)	0.0091	0.5635
- Splenectomy	13 (15.29%)	7 (18.92%)	6 (12.50%)		1.0000	11 (18.33%)	2 (8%)		1.0000
- Pancreatectomy	1 (1.18%)	0 (0%)	1 (2.08%)		1.0000	1 (1.67%)	0 (0%)		1.0000
- Cholecystectomy	48 (56.47%)	21 (56.76%)	27 (56.25%)		1.0000	37 (61.67%)	11 (44%)		1.0000
- Appendectomy	13 (15.29%)	8 (21.62%)	5 (10.42%)		1.0000	9 (15.00%)	4 (16%)		1.0000
- Bladder resection	1 (1.18%)	1 (2.70%)	0 (0%)		1.0000	1 (1.67%)	0 (0%)		1.0000
- Greater omentectomy	76 (89.41%)	34 (91.89%)	42 (87.50%)		1.0000	53 (88.33%)	23 (92%)		1.0000
- Lesser omentectomy	34 (40%)	19 (51.35%)	15 (31.25%)		1.0000	25 (41.67%)	9 (36%)		1.0000
- Liver resection	9 (10.59%)	6 (16.22%)	3 (6.25%)		1.0000	7 (11.67%)	2 (8%)		1.0000
- Stomach resection	6 (7.06%)	3 (8.11%)	3 (6.25%)		1.0000	5 (8.33%)	1 (4%)		1.0000
- Hysterectomy	60 (70.59%)	25 (67.57%)	35 (72.92%)		1.0000	45 (75.00%)	15 (60%)		1.0000
- Adnexectomy	65 (76.47%)	28 (75.68%)	37 (77.08%)		1.0000	43 (71.67%)	22 (88%)		1.0000
Length of hospital stay (day)	22.76 ± 12.29	24.16 ± 9.33	21.79 ± 14.37		1.0000	24.38 ± 13.39	19.08 ± 8.89		1.0000
Length of ICU stay (day)	5.23 ± 5.25	5.46 ± 5.28	5.06 ± 5.37		1.0000	5.72 ± 5.72	4.08 ± 3.99	0.0361	1.0000
Blood transfusion									
- Erythrocyte concentrates (unit)	1.46 ± 2.20	1.30 ± 2.26	1.46 ± 2.00		1.0000	1.60 ± 2.28	0.88 ± 1.54	0.0511	1.0000
- Fresh frozen plasma (unit)	3.45 ± 3.98	4.22 ± 5.03	2.73 ± 2.53		1.0000	3.73 ± 4.11	2.52 ± 3.14		1.0000
- 20:40:60 g Albumin	3:4:1 (3.53:4.71:1.18%)	0:1:0 (0:2.70:0%)	3:3:1 (6.25:6.25:2.08%)		1.0000	0:0:0 (0:0:0%)	3:4:1 (12:16:8%)		0.0015
Pleura punction	8 (9.41%)	4 (%)	4 (8.33%)		1.0000	5 (8.33%)	3 (12%)		1.0000
Dialysis	3 (3.53%)	1 (2.70%)	2 (4.17%)		1.0000	2 (3.33%)	1 (4%)		1.0000
Complication grade (Clavien–Dindo)					1.0000				1.0000
- II	37 (43.53%)	18 (48.65%)	19 (39.58%)			29 (48.33%)	8 (32%)		
- III	22 (25.88%)	12 (32.43%)	10 (20.83%)			14 (23.33%)	8 (32%)		
- IV	3 (3.53%)	1 (2.70%)	2 (4.17%)			2 (3.33%)	1 (4%)		
- V	2 (2.35%)	0 (0%)	2 (4.17%)			2 (3.33%)	0 (0%)		
Complications									
- Anastomotic insufficiency/leak	2 (2.35%)	2 (5.41%)	0 (0%)		1.0000	1 (1.67%)	1 (4%)		1.0000
- Pneumonia	7 (8.24%)	2 (5.41%)	5 (10.42%)		1.0000	6 (10.00%)	1 (4%)		1.0000
- Pulmonary embolism	4 (4.71%)	2 (5.41%)	2 (4.17%)		1.0000	4 (6.67%)	0 (0%)		1.0000
- Urinary tract infection	20 (23.53%)	10 (27.03%)	10 (20.83%)		1.0000	15 (25.00%)	5 (20%)		1.0000
- Pleural effusion	21 (24.71%)	11 (29.73%)	10 (20.83%)		1.0000	16 (26.67%)	5 (20%)		1.0000
- Fascial rupture	3 (3.53%)	1 (2.70%)	2 (4.17%)		1.0000	2 (3.33%)	1 (4%)		1.0000
- Renal insufficiency	7 (8.24%)	2 (5.41%)	5 (10.42%)		1.0000	3 (5.00%)	4 (16%)		1.0000
- Surgical site infection	10 (11.76%)	7 (18.92%)	3 (6.25%)		1.0000	9 (15.00%)	1 (4%)		1.0000
- Mortality (30 days)	2 (2.35%)	0 (0%)	2 (4.17%)		1.0000	2 (3.33%)	0 (0%)		1.0000
- Reoperation	12 (14.12%)	5 (13.51%)	7 (14.58%)		1.0000	8 (13.33%)	4 (16%)		1.0000
Median overall survival (month)	42.58	43.00	47.44		–	31.16	58.41		–
Median DSS (month)	45.24	45.24	47.44		–	41.92	58.41		–
Median RFS (month)	– ^1^	– ^1^	59.73		–	59.73	– ^1^		–

^1^ Not enough event occurred, median RFS not reached. Ana: anastomosis; ASA: American Society for Anesthesiologists; CC: Sugarbaker’s completeness of cytoreduction score; DSS: disease-specific survival; FIGO: Federation Internationale de Gynecolgie et d’Obstetrique; HGS: High-grade serous; HIPEC: Hyperthermic Intraperitoneal Chemotherapy; ICU: intensive care unit; met: metastasis; NACT: neoadjuvant chemotherapy; perit: peritonectomy; RFS: recurrence-free survival; TAX: paclitaxel.

**Table 2 cancers-14-03301-t002:** Results of uni- and multivariate survival models.

Clinical Characteristics	*DSS*		*OS*		RFS	
	Univariate *p*-Value	Multivariate *p*-Value	Univariate *p*-Value	Multivariate *p*-Value	Univariate *p*-Value	Multivariate *p*-Value
Age (years)	0.0307	0.0113	0.0088	0.0416	0.6850	0.4184
Body mass index (kg/m^2^)	0.0230	0.0004	0.0301	0.0044	0.6630	0.7968
ASA score	<0.0001	<0.0001	0.2821	0.1319	<0.0001	<0.0001
FIGO Stage	0.4550	<0.0001	0.3085	0.2154	<0.0001	<0.0001
CC score	0.0921	0.1617	0.0888	0.2351	0.3446	0.1521
Peritoneal carcinomatosis index	0.0724	< 0.0001	0.1040	0.0062	0.7090	0.1075
Type of HIPEC (upfront vs. interval)	0.3670	< 0.0001	0.2410	0.0016	0.8240	0.1865
Duration of HIPEC (60 vs. 90 min)	0.0538	0.0030	0.0330	0.0012	0.2070	0.0959
Complication grade (Clavien–Dindo)	0.1340	– ^1^	0.0018	0.0041	<0.0001	<0.0001
Any incomplete tumor removal surgery prior CRS + HIPEC	0.4300	0.0655	0.4290	0.2552	0.8740	0.9810

^1^ Computation of Type II Analysis of Deviance Table *p*-value was not feasible. ASA: American Society for Anesthesiologists; CC: Sugarbaker’s completeness of cytoreduction score; DSS: disease-specific survival; FIGO: Federation Internationale de Gynecolgie et d’Obstetrique; HIPEC: hyperthermic intraperitoneal chemotherapy; RFS: recurrence-free survival; OS: overall survival.

## Data Availability

The data presented in this study are available on request from the corresponding author.

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
