# Peer review of "Prolonged Exposition with Hyperthermic Intraperitoneal Chemotherapy (HIPEC) May Provide Survival Benefit after Cytoreductive Surgery (CRS) in Advanced Primary Epithelial Ovarian, Fallopian Tube, and Primary Peritoneal Cancer"

_cancers, 2022, doi:10.3390/cancers14143301_

Round 1

Reviewer 1 Report

Dear editor

First of all, I would like to thank CANCERS for the possibility of being able to review the work entitled: “Prolonged Exposition with Hyperthermic Intraperitoneal Chemotherapy (HIPEC) May Provide Survival Benefit after Cytoreductive Surgery (CRS) in Advanced Primary Epithelial Ovarian Cancer, Fallopian Tube and Primary Peritoneal Cancer".

The paper by Micklos et al studies 2 clear objectives: the difference in survival rates between 2 clinical scenarios (up-front and IDS) and, on the other hand, the difference between 2 different times of exposure to HIPEC treatment.

This is a retrospective analysis of a series of 87 patients with IIIB-IV ovarian cancer operated on over a 10-year period.

The paper is similar to the one previously published and is part of the bibliography (reference 36). Authors should specify whether the patients in one publication are also part of the other publication and specify this in the text of the new version of the manuscript. This is an important editorial point.

The scientific language is clear and the paper is easy to read.

However, I would like to make some considerations that must be taken into account:

1- A suggestion: in my opinion, the authors should have excluded 2 patients described in lines 111-115: “One 111 patient received gemcitabine (2000 mg/m2 of body surface area) HIPEC due to intolerance 112 of cisplatin during preoperative systemic chemotherapy, and another patient received ox-113 aliplatin (300 mg/m2 of body surface area) HIPEC with intravenous fluorouracil (400 114 mg/m2) and folinic acid (20 mg/m2)”.

2- Patients for primary surgery with liver metastases have been included. Lines 189-190 specify: “FIGO stage 188 IV (crude p = 0.0066) and the presence of hepatic and/or splenic parenchymal metastasis 189 (crude p = 0.0387) was more common in the upfront HIPEC group”. In these patients, were NACT not considered? Please clarify.

3- The survival results are similar between the up-front group and the IDS group, and that less aggressive surgeries were performed, with fewer ileostomies in patients treated with NACT, would this be a good reason to reinforce the use of NACT after of these results?

4- What were the reasons for changing the HIPEC treatment protocol in 2018; van Driel's study? please clarify

5- It would be necessary to clarify the concept of “incomplete tumor removal” (Line 240). In patients with a first surgery (with or without NACT), only biopsy should be considered as permitted surgery before CRS. Other types of surgery, including incomplete or suboptimal procedures, would change the scenario, and should not be included as up-front surgeries or IDS. This is an important issue of the study that must be clarified to be sure that we are talking about up-front or IDS, or something else.

Reviewer 2 Report

Dr. Acs and colleagues from Prof. Piso's team analyzed retrospectively their single-center experience with HIPEC for epithelial GYN cancer (n=87) over a 10-year period. Main finding was a significant benefit of longer HIPEC duration (90 vs 60min) for OS. 

The study is timely and of interest, and the paper is well written. Several questions/suggestions should be answered/considered:

- This is a retrospective series over 10 years period of time. Retrospective nature should be clearly stated and changes of care/team etc. need to be detailed as they are potential confounders.

- There are two main comparisons (interval/upfront; duration HIPEC) making already 4 subgroups + another subgroup of incomplete surgery. This is not only statistically too much for n=87. Main problems hee are statistical adjustment for multiple group comparison and type II error. A descriptive presentation of results with mutivariable analysis should be the best way.

- Please provide STROBE statement.

- Please provide n of patients below the remaining survival curves.

- The authors used Cis/Dox as HIPEC regimen. Do you still use it after 3 RCTs using only Cis. If still combination, please justify! If not, does the discussion about duration change for Cis as monotherapy?

- Can information be provided on post-OP chemotherapy as obvious confounder for OS?

Congratulations for an excellent work!

Round 2

Reviewer 1 Report

Dear Editor

I believe that the modifications and clarifications that the authors have made in the new version of the manuscript submitted for publication are sufficient. They have also made an important effort recalculating all the statistical data.

I am satisfied with these modifications.

Reviewer 2 Report

Thanks to the authors to provide a careful revision adressing the points raised by the reviewers.